# TACI Mutations in Primary Antibody Deficiencies: A Nationwide Study in Greece

**DOI:** 10.3390/medicina57080827

**Published:** 2021-08-16

**Authors:** Ioannis Kakkas, Gerasimina Tsinti, Fani Kalala, Evangelia Farmaki, Alexandra Kourakli, Androniki Kapousouzi, Maria Dimou, Vassiliki Kalaitzidou, Eirini Sevdali, Athanasia-Marina Peristeri, Georgia Tsiouma, Peristera Patiou, Eleni Papadimitriou, Theodoros P. Vassilakopoulos, Panayiotis Panayiotidis, Anna Kioumi, Argiris Symeonidis, Matthaios Speletas

**Affiliations:** 1Immunology and Histocompatibility Department, “Evaggelismos” General Hospital, 106 76 Athens, Greece; ioankakkas@gmail.com; 2Department of Immunology & Histocompatibility, Faculty of Medicine, University of Thessaly, 415 00 Larissa, Greece; biomintsi@gmail.com (G.T.); nikikapousouzi@gmail.com (A.K.); esevdali@hotmail.gr (E.S.); nancy.p12@hotmail.com (A.-M.P.); 3Hematology Department, Henry Dunant Hospital, 115 26 Athens, Greece; fanikalala@hotmail.com; 4Pediatric Immunology and Rheumatology Referral Center, First Department of Pediatrics, Aristotle University of Thessaloniki, 541 24 Thessaloniki, Greece; farmakg@auth.gr (E.F.); e.papadimitriu@gmail.com (E.P.); 5Department of Internal Medicine, Hematology Division, Medical School—University Hospital, University of Patras, 265 04 Patras, Greece; akourakli@hotmail.com (A.K.); peripatiou@yahoo.gr (P.P.); argiris.symeonidis@yahoo.gr (A.S.); 6Hematology Section, First Department of Propedeutic Internal Medicine, “Laikon” General Hospital, National and Kapodistrian University of Athens, 157 72 Athens, Greece; msdimou@gmail.com (M.D.); ppanayi@med.uoa.gr (P.P.); 7Department of Hematology, Papageorgiou General Hospital, 564 29 Thessaloniki, Greece; kalaitzidou_b@yahoo.gr (V.K.); annakioumi@gmail.com (A.K.); 8ENT Department, “Achillopoulion” General Hospital of Volos, 382 21 Volos, Greece; gtsiouma@yahoo.gr; 9Department of Haematology and Bone Marrow Transplantation, “Laikon” General Hospital, National and Kapodistrian University of Athens, 157 72 Athens, Greece; theopvass@hotmail.com

**Keywords:** TACI, CVID, autoimmune cytopenias, benign lymphoproliferation

## Abstract

*Background and objectives*: Monoallelic (heterozygous) or biallelic (homozygous or compound heterozygous) TACI mutations have been reported as the most common genetic defects in patients with common variable immunodeficiency (CVID), which is the most common clinically significant primary immunodeficiency in humans. The aim of our study was to evaluate the prevalence and any correlations of TACI defects in Greek patients with primary antibody deficiencies. *Materials and Methods*: 117 patients (male/female: 53/64) with CVID (110) and a combined IgA and IgG subclass deficiency (7) with a CVID-like clinical phenotype were enrolled in the study. Genomic DNA was extracted from peripheral blood and the molecular analysis of the *TACI* gene was performed by PCR (Polymerase Chain Reaction) and sequencing of all 5 exons, including exon–intron boundaries. *Results*: Seventeen patients (14.5%) displayed TACI defects, four (23.5%) carried combined heterozygous mutations and 13 (76.5%) carried single heterozygous mutations. The most frequently detected mutation was C104R (58.8%), followed by I87N (23.5%) and A181E (11.8%), while R20C, C62Y, P151L, K188M and E236X mutations were present in only one patient each. Patients with TACI defects were more frequently male (*p* = 0.011) and displayed a benign lymphoproliferation (splenomegaly and lymph node enlargement, *p* = 0.047 and *p* = 0.002, respectively), had a history of tonsillectomy (*p* = 0.015) and adenoidectomy (*p* = 0.031) and more frequently exhibited autoimmune cytopenias (*p* = 0.046). *Conclusions*: Considering that accumulating evidence suggests several CVID patients have a complex rather than a monogenic inheritance, our data further support the notion that TACI mutations, particularly as monoallelic defects, should be primarily considered as susceptibility co-factors and/or modifiers of primary antibody deficiencies.

## 1. Introduction

Primary antibody deficiencies (PAD) represent the most common types of primary immunodeficiencies in humans [1]. Among them, common variable immunodeficiency (CVID) is the most frequent symptomatic disorder worldwide, with an estimated prevalence of approximately 1:25,000 in the general population [1,2,3]. In fact, CVID represents a heterogeneous group of diseases with common clinical and laboratory findings, including hypogammaglobulinemia (IgG in all, IgA in 70–80% and IgM in approximately 50% of affected patients), weak or absent responses against polysaccharide (mainly) and protein antigens and usually very low isotype-switched memory B cell counts in the periphery [2,3]. The range of CVID clinical manifestations is broad, with infections being the most common. However, affected patients display a high frequency of benign lymphoproliferation (splenomegaly, lymph node enlargement and lymph infiltrates in the gastrointestinal track), recurrent attacks of autoimmune cytopenias, granulomatous disease as well as an enhanced risk of malignancy, especially lymphomas and gastrointestinal cancer [2,3,4].

Until now, 14 distinct genetic defects causing CVID have been described in the Online Mendelian Inheritance in Man (OMIM) database (https://www.omim.org, accessed 11 January 2021). Among them, TNFRSF13B/TACI mutations represent the most common molecular defects related to the disease [5]. Interestingly, TNFRSF13B/TACI mutations have also been reported in patients with IgA deficiency (IgAD), namely the most common immunodeficiency in the Western world, with a prevalence of approximately 1 in 700 individuals [3,6]. However, most IgAD patients are asymptomatic, and only one third of them may display recurrent infections and/or autoimmunity [3,6]. There is increased evidence that patients with combined IgA and IgG subclass (IgGs) deficiencies (usually IgG2 or IgG4) are more prone to developing CVID-like manifestations [3]. In this context, we have recently reported that patients with selective IgGs deficiency (especially of IgG4) also display an increased incidence of TNFRSF13B/TACI defects [7]. Whether these defects are a predisposition for clinical manifestations in patients with combined IgA and IgGs deficiencies is still unclear.

It is worth noting that heterozygous TACI defects have also been detected in patients’ relatives who displayed no disease, as well as in healthy individuals [8,9]. Moreover, by analyzing the contribution of TACI defects to CVID pathogenesis and phenotype in 564 patients, Salzer et al. demonstrated that two mutations, namely p.C104R and p.A181E, are the most common and have a more profound effect on disease phenotype [10]. Furthermore, Salzer et al. reported that the presence of TACI mutations is mainly correlated with an increased susceptibility to splenomegaly and autoimmunity in CVID [10]. However, considering that the TACI gene is very polymorphic, the incidence and the clinical significance of its alterations may be different in other ethnic groups.

In this context, the aim of our study was to clarify the prevalence of TACI defects in Greek PAD patients, particularly those with CVID and combined IgA and IGs deficiency, in order to evaluate any potential correlations with their clinical phenotype.

## 2. Materials and Methods

### 2.1. Patients Characteristics

One-hundred and seventeen (117) patients (male/female: 53/64) with PAD from 109 families were enrolled in the study. The majority of patients (110, 94.0%) fulfilled the diagnostic criteria of CVID (low levels of serum IgG, IgA and/or IgM greater than two standard deviations from the normal mean, complete absence or poor response to polysaccharide vaccines and an exclusion of other defined causes of hypogammaglobulinemia), and seven patients (6.0%) exhibited combined IgA and IgGs deficiencies with CVID-like clinical manifestations. All patients of the latter group displayed IgA and IgG4 deficiencies, two of them also had an IgG2 deficiency, two had an IgG1 deficiency and one had an IgG3 deficiency.

The median age of patients at the time of diagnosis was 38 years (range: 4–70) and at the time of analysis was 44 years (range: 15–76). The most common clinical manifestation (108 patients, 93.2%) was infection (mainly respiratory), while 60 patients (51.3%) had splenomegaly, 47 (40.2%) lymphadenopathy, 26 (22.2%) hepatomegaly and 29 patients (24.7%) displayed various autoimmune cytopenias (autoimmune hemolytic anemia, autoimmune thrombocytopenia or Evans syndrome). Additionally, 11 patients (9.4%) exhibited a granulomatous disease (located on lymph nodes, spleen or skin), 35 (29.9%) had a history of atopy (commonly in antibiotic drugs), 16 (13.7%) had a history of tonsillectomy and nine (7.6%) a history of adenoidectomy. A splenectomy was performed in 13 patients (11.1%) for diagnostic (4, 30.8%) or therapeutic purposes (hypersplenism, resistant autoimmune manifestations, spontaneous rupture) (9, 69.2%). Twenty-two patients (18.8%) developed different types of neoplasia, including lymphomas (10), acute lymphoblastic leukemia (2), breast cancer (3), gastric cancer (2), colon cancer (1), uterus cancer (2), cervical cancer (1), thyroid cancer (1) and fibrosarcoma (1). Interestingly, three patients with lymphoma relapsed with another distinct type (a patient who initially developed Burkitt lymphoma relapsed with Hodgkin disease, a second one who was initially diagnosed with marginal zone lymphoma, relapsed with diffuse large B cell lymphoma, and a third patient who initially had developed Hodgkin disease relapsed with diffuse large B cell lymphoma) and another patient who initially had developed cervical cancer, progressed to gastric cancer 13 years later. At the time of analysis, 104 patients (88.9%) were subjected to immunoglobulin replacement treatment (intravenous, subcutaneous or facilitated subcutaneous); among the 13 remaining, one patient with CVID refused treatment, two patients with combined IgA and IgGs deficiency were receiving prophylactic antibiotic treatment, and 10 were still at the baseline phase of their investigation. An overview of patient characteristics is presented in Table 1.

Written informed consent was obtained from each individual or an accompanying relative, in the case of patients from whom consent was not legally applicable (e.g., children). The study was approved by the ethical committee of the University Hospital of Larissa (17831/18-4-2018) and the ethical committee of the Faculty of Medicine, University of Thessaly (10/12-2-2019) and was carried out in accordance with the principles of the Helsinki Declaration.

### 2.2. Molecular Analysis

Genomic DNA was extracted from peripheral blood using the QIAamp DNA Blood Mini Kit (Qiagen Ltd., Crawley, UK), according to manufacturer’s instructions. Afterward, a PCR amplification of all five exons (including exon–intron boundaries) of the TACI gene was performed as described [7].

### 2.3. Statistical Analysis

Quantitative variables are presented with mean and standard deviation (SD). Qualitative variables are presented with absolute and relative frequencies. Variables were first tested for normality with the Kolmogorov–Smirnov test. If the normality assumption was satisfied for the comparison of means between two groups, Student’s t-test was used. For the comparisons of the proportions chi-square and Fisher’s exact tests with Yates correction were computed. A series of stepwise multiple logistic regression analyses (*p* for removal was set at 0.10 and *p* for entry was set at 0.05) were conducted to investigate whether TACI mutations were independently associated with specific outcomes. Adjusted odds ratios with 95% confidence intervals for TACI mutation were computed from the results of the logistic regression analyses. All *p* values reported are two-tailed. Statistical significance was set at 0.05 and analyses were conducted using SPSS statistical software (version 23.0).

## 3. Results

### 3.1. Prevalence of TACI Mutations in the Patients of the Study

Seventeen patients (14.5%) displayed TACI pathogenic defects, including 15 out of 110 patients with CVID (13.6%) and two out of seven patients with combined IgAD and IgGsD (28.6%). None of them carried a homozygous defect; four patients (23.5%) carried combined heterozygous mutations and 12 (76.5%) single heterozygous TACI mutations (Table 2). The most frequently detected mutation was the C104R (rs34557412) which was present in 10 patients (58.8%), followed by the I87N allele (rs72553877) present in four patients (23.5%) and by the A181E allele (rs72553883) present in two patients (11.8%). The following mutated alleles were present in only one patient: R20C (rs200013015), C62Y (rs1410473109), P151L (rs200037919), K188M (rs74811083) and E236X (rs201021960). In addition, the C014R defect was present in all four combined heterozygous mutations and the A181E in two of them.

Considering the common polymorphisms of the TACI gene (intronic, silent and missense mutations), the most commonly detected in our cohort was the silent mutation rs8072293 (p.T27=, allele frequency 68.8%), followed by the intronic polymorphism rs2274892 (g.24625A>C, allele frequency 43.2%), the silent mutation rs11078355 (p.S277=, allele frequency 41.0%), the intronic polymorphisms rs11652843 (g.33402T>G, allele frequency 31.6%) and rs11652811 (g.33482T>C, allele frequency 31.6%), the silent mutation rs35062843 (p.97P=, allele frequency 4.3%) and the missense mutations (polymorphisms) rs56063729 (p.V220A, allele frequency 3.0%) and rs34562254 (p.P251L, allele frequency 10.7%). Finally, the distribution of exonic and intronic polymorphisms was similar in those patients with or without the TACI rare defects, except for the silent mutation rs11078355 (p.S227=), which was expressed less frequently in patients with TACI defects (Table 2).

### 3.2. Associations of TACI Defects with Clinical Manifestations of the Patients of the Study

As presented in detail in Table 3, males were more likely to have a TACI mutation as compared to females (*p* = 0.011). Moreover, patients with TACI molecular defects were more likely to have lymphoproliferation (*p* = 0.004), especially splenomegaly (*p* = 0.047) and lymphadenopathy (*p* = 0.002), as well as higher incidence of being subjected to tonsillectomy (*p* = 0.015) and adenoidectomy (*p* = 0.031). Interestingly, almost half of patients with TACI defects in our cohort exhibited recurrent attacks of autoimmune hematologic complications (autoimmune hemolytic anemia, autoimmune thrombocytopenia or Evans syndrome), and the difference compared to patients with no mutations was also statistically significant (Table 3). Notably, the most prevalent TACI defect was that of C104R (6 out of 10 patients, 60%). Conversely, no significant correlations of TACI mutations with the presence and the type of infections (upper respiratory, *p* = 0.865; lower respiratory, *p* = 0.254; urinary, *p* = 0.897; gastrointestinal, *p* = 0.727; other locations including skin, CNS, etc., *p* = 0.792) were observed. Similarly, no significant correlations of TACI molecular defects with the presence of granulomatous disease and atopy, or the emergence of neoplasia were also found (Table 3).

Following the significant association of TACI pathogenic defects with benign lymphoproliferation (splenomegaly, lymphadenopathy), history of adenoidectomy and tonsillectomy as well as the presence of hematological autoimmunity in the univariate analysis (Table 3), a series of stepwise multiple logistic regression analyses were performed. These were to investigate whether the presence of TACI mutations was independently associated with the aforementioned outcomes. It was found that patients with TACI defects had 3.34 times greater odds ratio for autoimmune cytopenias (95% confidence interval [CI]: 1.15–9.72, *p* = 0.027), 13.33 times greater odds ratio for any type of lymphoproliferation (95% CI: 1.70–104.40, *p* = 0.014) and more specifically 3.59 times greater for splenomegaly (95% CI: 1.02–11.79, *p* = 0.036), 6.21 times greater for lymphadenopathy (95% CI: 1.88–20.52, *p* = 0.004), 5.84 times greater for adenoidectomy (95% CI: 1.39–4.59, *p* = 0.016) and 4.91 times greater for tonsillectomy (95% CI: 1.49–16.14, *p* = 0.009).

## 4. Discussion

In this study, we identified TACI mutations in 14.5% of PAD patients, and their presence was significantly associated with an increased incidence of benign lymphoproliferation (splenectomy, lymphadenectomy and medical history of tonsillectomy and/or adenoidectomy), as well as a high prevalence of autoimmune cytopenias. Thus, our results in Greek patients with PAD were rather similar to those observed in a larger study published by Salzer et al. as mentioned above [10]. Moreover, our data further support the notion that TACI defects have a remarkable impact on autoimmunity prevalence, especially the C104R variant, in patients with CVID [11].

TACI is a receptor that binds two ligands: a proliferation inducing ligand (APRIL) and a B-cell activating factor (BAFF). It is preferentially expressed on the marginal zone and isotype-switched memory B cells and plasma cells [12]. TACI monoallelic (heterozygous) or biallelic (homozygous or compound heterozygous) defects have been reported in 5–20% of patients with CVID and IgAD in several geographic areas globally [10,13,14,15]. Our results clearly support this finding.

Although TACI-knockout mice did not display hypogammaglobulinemia, the absence of TACI signaling resulted in a prominent autoimmune phenotype and signs of lymphoproliferation [12,16]. Interestingly, Zhang et al. and Salzer et al. were the first to observe a higher prevalence of autoimmune cytopenias and benign lymphoproliferation in CVID patients [10,17], and our study further supports these associations. However, TACI defects have also been reported in patients with sarcoidosis and tonsillar hypertrophy [18], but not in patients with systemic lupus erythematosus or other autoimmune disorders [19]. On the other hand, although heterozygous TACI defects have been demonstrated to be pathogenic in “in vitro” studies [20], such defects have been observed in both relatives of PAD patients without overt disease and healthy individuals [7,9,18]. Therefore, considering that accumulating evidence suggests several PAD patients have a complex rather than a monogenic inheritance [5,21], the monoallelic TACI alterations should be primarily considered as susceptibility factors and/or modifiers of PAD.

In our study, we have demonstrated a high prevalence of TACI defects among Greek patients with combined IgA and IgGs deficiencies with a CVID-like phenotype. To the best of our knowledge this is the first study in the literature describing such an association. Obviously, further studies with a larger cohort of similar patients are necessary to confirm this finding.

## 5. Conclusions

In conclusion, our results confirmed previous studies that TACI defects are present in approximately 15% of PAD patients, and that they are associated with a benign lymphoproliferation and autoimmune cytopenias. Taking into consideration that monoallelic TACI defects are also present in non-PAD individuals, further studies analyzing a broad spectrum of causative genes are necessary in order to highlight the emergence of PAD in humans.

## Figures and Tables

**Table 1 medicina-57-00827-t001:** An overview of demographic and clinical characteristics of the patients of the study.

	Total	CVID	Combined IgAD & IgGsD
No	117	110	7
Sex (male/female)	53/64	48/62	5/2
Age at analysis (mean ± SD)	44.1 ± 15.9	44.8 ± 15.7	32.0 ± 13.9
Age at diagnosis (mean ± SD)	36.0 ± 15.6	36.4 ± 15.6	28.4 ± 14.2
Age at disease onset (mean ± SD)	24.7 ± 15.4	25.6 ± 15.3	11.1 ± 10.7
Lymphoproliferation (no, %)	70 (59.8)	67 (60.1)	3 (42.9)
Splenomegaly (no, %)	60 (51.3)	57 (51.2)	3 (42.9)
Lymphadenopathy (no, %)	47 (40.2)	45 (40.9)	2 (28.6)
Intestine infiltrates (no, %)	10 (8.5)	9 (8.2)	1 (14.3)
Hepatomegaly (no, %)	26 (22.2)	25 (25.0)	1 (14.3)
Infections (no, %)	109 (93.2)	102 (92.7)	7 (100.0)
Upper respiratory (no, %)	98 (83.8)	92 (83.6)	6 (85.7)
Lower respiratory (no, %)	73 (62.4)	70 (63.6)	3 (42.9)
Gastrointestinal (no, %)	28 (23.9)	27 (24.5)	1 (14.3)
Urinary (no, %)	29 (24.8)	29 (26.4)	0 (0)
Others ^ (no, %)	27 (23.1)	27 (24.5)	0 (0)
Bronchiectasis (no, %)	27 (23.1)	26 (23.6)	1 (14.3)
Granulomatous disease (no, %)	11 (9.4)	10 (9.1)	1 (14.3)
Autoimmune manifestations (no, %)	67 (57.3)	61 (55.5)	6 (85.7)
Thyroid disease (no, %)	27 (23.1)	24 (21.8)	3 (42.9)
AHA and/or ATP and/or Evans syndrome (no, %)	29 (24.8)	26 (23.6)	3 (42.9)
Others ^#^ (no, %)	25 (21.4)	22 (20.0)	3 (42.9)
Atopy (no, %)	34 (29.1)	33 (30.0)	1 (14.3)
Splenectomy (no, %)	13 (11.1)	13 (11.8)	0 (0)
Adenoidectomy (no, %)	9 (7.7)	7 (6.4)	2 (28.6)
Tonsillectomy (no, %)	16 (13.7)	13 (11.8)	3 (42.3)
Neoplasia (no, %)	22 (18.8)	22 (20.0)	0 (0)
Other complications * (no, %)	10 (8.5)	10 (9.1)	0 (0)
Under replacement treatment (no, %)	105 (89.7)	100 (90.9)	5 (71.4)

**Abbreviations**: AHA, autoimmune hemolytic anemia; ATP, autoimmune thrombocytopenic purpura; CVID, Common Variable Immunodeficiency; SD, standard deviation. ^ They include skin infections, peritonitis, sepsis or meningitis. # They include psoriasis, vitiligo, pernicious anemia, myelitis, multiple sclerosis, Raynaud syndrome, lupus erythematosus, primary biliary cirrhosis. * They include nodular hyperplasia of the liver, cirrhosis and hypersplenism, portal vein thrombosis, malabsorption, spleen rupture, renal insufficiency.

**Table 2 medicina-57-00827-t002:** *TACI* defects, exonic and intronic polymorphisms in the patients of the study.

	TACI defects			
	Total	No	Yes	
	117 pts	100 pts	17 pts	
	No (%)	No (%)	No (%)	*p*
**A. CVID causative/modifier mutations**				
Homozygous	0 (0.0)		0 (0.0)	
Combined heterozygous	4 (3.4)		4 (23.5)	
Heterozygous	13 (11.1)		13 (76.5)	
rs34557412(p.C104R)	10 (8.5)		10 (58.8)	
rs72553883 (p.A181E)	2 (1.7)		2 (11.8)	
rs200013015 (R20C)	1 (0.9)		1 (5.9)	
rs1410473109 (C62Y)	1 (0.9)		1 (5.9)	
rs72553877 (p.I87N)	4 (3.4)		4 (23.5)	
rs200037919 (p.P151L)	1 (0.9)		1 (5.9)	
rs74811083 (p.K188M)	1 (0.9)		1 (5.9)	
rs201021960 (p.E236X)	1 (0.9)		1 (5.9)	
**B. TACI exonic and intronic polymorphisms (No, allele frequencies %)**				
rs8072293 (p.T27=)	161, 68.8	135, 67.5	26, 76.5	0.325
rs35062843 (p.97P=)	10, 4.3	10, 5.0	0, 0.0	0.365
rs56063729 (p.V220A)	7, 3.0	7, 3.5	0, 0.0	0.573
rs11078355 (p.S277=)	96, 41.0	88, 44.0	8, 23.5	0.039
rs34562254 (p.P251L)	25, 10.7	23, 11.5	2, 5.9	0.496
rs2274892 (g.24625A>C)	101, 43.2	91, 45.5	10, 29.4	0.118
rs11652843 (g.33402T>G)	74, 31.6	67, 33.5	7, 20.6	0.194
rs11652811 (g.33482T>C)	74, 31.6	67, 33.5	7, 20.6	0.194

Numbers in bold represent the significant differences.

**Table 3 medicina-57-00827-t003:** Association of TACI defects with clinical characteristics of the patients of the study.

Clinical Characteristics	TACI Molecular Defects		
	No	Yes	*p*
	100 pts	17 pts	
	No (%)	No (%)	
Diagnosis			0.593
CVID	95 (86.4)	15 (13.6)	
Combined IgAD and IgGsD	5 (71.4)	2 (28.6)	
Sex			**0.011**
Male	40 (75.5)	13 (24.5)	
Female	60 (93.7)	4 (6.3)	
Age of disease onset, mean (SD)	25.2 (15.5)	21.9 (15.1)	0.429
Age at diagnosis, mean (SD)	36.4 (15.4)	33.4 (17.1)	0.460
Lymphoproliferation	54 (54.0)	16 (94.1)	**0.004**
Splenomegaly	47 (47.0)	13 (76.5)	**0.047**
Lymphadenopathy	34 (34.0)	13 (76.5)	**0.002**
Intestine lymph infiltrates	9 (9.0)	1 (5.9)	0.670
Hepatomegaly	21 (21.0)	5 (29.1)	0.649
Infections	93 (93.0)	16 (94.1)	0.866
Bronchiectasis	25 (25.0)	2 (11.8)	0.376
Granulomatous disease	10 (10.0)	1 (5.9)	0.929
Autoimmune manifestations	56 (56.0)	11 (64.7)	0.685
Thyroid disease	25 (25.0)	2 (11.7)	0.376
AHA and/or ATP and/or Evans syndrome	21 (21.0)	8 (47.1)	**0.046**
Others *	21 (21.0)	4 (23.5)	0.814
Atopy	28 (28.0)	6 (35.3)	0.746
Splenectomy	10 (10.0)	3 (17.6)	0.610
Adenoidectomy	5 (5.0)	4 (23.5)	**0.031**
Tonsillectomy	10 (10.0)	6 (35.3)	**0.015**
Neoplasia	19 (19.0)	3 (17.6)	0.895
Other complications	8 (8.0)	2 (11.7)	0.965

* Other autoimmune manifestations: autoimmune hepatitis, autoimmune neutropenia, eczema, lupus, myelitis, pernicious anemia, primary biliary cirrhosis, psoriasis, Raynaud syndrome, vasculitis, vitiligo. Numbers in bold represent the significant differences.

## Data Availability

All data being analyzed in this manuscript are available upon request to the corresponding author.

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
