# Peer review of "TACI Mutations in Primary Antibody Deficiencies: A Nationwide Study in Greece"

_medicina, 2021, doi:10.3390/medicina57080827_

Round 1

Reviewer 1 Report

The study of Kakkas I. and coll. “TACI mutations in primary antibody deficiencies. A nation-wide study in Greece” describes the presence and the frequency of TACI mutations in a cohort of patients affected by primary antibody deficiency (PAD), mainly CVID and a small group of patients carrying combined IgA and IgG subclass deficiency.

Overall, the results obtained confirm the prevalence and the relative frequency of specific TACI mutations in PAD patients and the association of TACI defects with lymphoproliferation and autoimmunity.

The study is properly designed and results are clearly presented. Only a few of point need to be improved.

The association of TACI defects with autoimmunity should be better discussed, including analyzing whether a specific TACI mutation (C104R?) is prevalent in these patients (1).

Editing of the tables should be improved in terms of readability and, most importantly, each individual table should be edited on a single page.

In table 1 the “others” category has a “special character” (^,  #, *) indicating a note. Please correct or add the note.

  1. Barroeta Seijas AB, Graziani S, Cancrini C et al. Int J Immunopathol Pharmacol. Apr-Jun 2012;25(2):407-14. doi: 10.1177/039463201202500210.

Author Response

Considering the comments and recommendations of the Reviewer #1:

The association of TACI defects with autoimmunity should be better discussed, including analyzing whether a specific TACI mutation (C104R?) is prevalent in these patients (1).

Barroeta Seijas AB, Graziani S, Cancrini C et al. Int J Immunopathol Pharmacol. Apr-Jun 2012;25(2):407-14. doi: 10.1177/039463201202500210

We have added the appropriate data in lines 186-187 (in Results section) and discussed it appropriately in lines 215-217, also adding the recommended Reference.

Editing of the tables should be improved in terms of readability and, most importantly, each individual table should be edited on a single page”.

We have appropriately modified our Tables according to the reviewer’s recommendation

In table 1 the “others” category has a “special character” (^,  #, *) indicating a note. Please correct or add the note

We have appropriately added the missing information (as Abbreviations of Table 1).

Reviewer 2 Report

Kakkas et al proved that several CVID patients have a complex rather than a monogenic inheritance, our data further supports the notion that TACI mutations, particularly as monoallelic defects, should be primarily considered as susceptibility co-factors and/or modifiers of primary antibody deficiencies. The paper is nicely presented.

Author Response

We are grateful for the reviewer's positive comments concerning our work.

Reviewer 3 Report

The article is devoted to the assessment of the prevalence and various correlations of TACI defects in 117 Greek patients with CVID (110) and combined deficiency of the IgA and IgG subclass (7) with a clinical phenotype similar to CVID. The authors showed that TACI mutations, especially as monoallelic defects, should be considered primarily as susceptibility cofactors and / or modifiers of primary antibody deficiency. In my opinion, the work was done at a good methodological level. The applied methods are adequate for the task at hand. The conclusions are consistent with the results obtained. In my opinion, given the undoubted importance of this work for medicine.

Author Response

(The authors gave the same response as above.)
